# Oxidative Stress Model of Lipopolysaccharide-Challenge in Piglets of Wuzhishan Miniature Pig

**DOI:** 10.3390/vetsci12080694

**Published:** 2025-07-24

**Authors:** Ruiying Bao, Pingfei Qiu, Yanrong Hu, Junpu Chen, Xiaochun Li, Qin Wang, Yongqiang Li, Huiyu Shi, Haiwen Zhang, Xuemei Wang

**Affiliations:** Animal Nutrition and Feed Laboratory, School of Tropical Agriculture and Forestry, Hainan University, Danzhou 571737, China; baoruiying524@163.com (R.B.); qpf773@163.com (P.Q.); hhyryh@163.com (Y.H.); chenjunpu8@gmail.com (J.C.); l_x_chun@163.com (X.L.); wangqin20210327@163.com (Q.W.); 15595790518@163.com (Y.L.); shihuiyu2017@163.com (H.S.); hwzhang@hainanu.edu.cn (H.Z.)

**Keywords:** oxidative stress, microbiota, metabolites, piglets

## Abstract

Oxidative stress is a critical factor affecting porcine health and productivity; rendering its study essential for advancements in animal science and agriculture. Miniature pig breeds, such as the Wuzhishan pig, exhibit substantial potential as medical models for physiological or pathological research. Lipopolysaccharide (LPS), a unique component of the outer membrane of Gram-negative bacteria, possesses immune-activating properties and is widely used to establish inflammation models due to its potent pro-inflammatory effects. This study aimed to characterize the LPS-induced oxidative stress model in Wuzhishan pigs. Results showed that LPS-challenge induced oxidative stress, leading to reduced body weight gain, impaired antioxidant capacity, elevated inflammatory levels, histopathological damage in intestinal and immune organs, altered metabolic profiles and microbiota dysregulation in piglets.

## 1. Introduction

The Wuzhishan miniature pig (WMP), also known as the “five-legged pig” due to its elongated snout, which is colloquially likened to a fifth appendage, is one of China’s most renowned miniature pig breeds. This miniature pig holds significant potential as a medical model for physiological or pathological studies [1,2]. However, this breed is rarely reported in oxidative stress (OS) or relevant studies.

Under physiological conditions, reactive oxygen species (ROS) and free radicals are maintained in dynamic homeostasis, but these molecules cause damage in vivo with overproduction when an imbalance occurs between oxidant and anti-oxidant systems. OS in pigs could be triggered by diverse stressors, including ammonia exposure in the lungs, early weaning, and polyketide mycotoxin exposure [3,4,5]. In addition, lipopolysaccharide (LPS) exposure induces significant OS in chickens, and the extent of OS was involved in LPS doses, target tissues and dietary ingredients [6]. LPS, a key component of the Gram-negative bacterial outer membrane, is a potent pro-inflammatory agent widely utilized to establish inflammation models, particularly due to its ability to robustly activate the immune system [7]. Its pro-inflammatory effects are primarily mediated through the activation of the NF-κB pathway. This activation leads to the upregulation of pro-inflammatory cytokines and a concomitant reduction in antioxidant enzyme activity, thereby triggering a well-defined cascade of oxidative stress [8,9]. This established inflammatory-oxidative stress pathway has been validated in various models, including sepsis and tissue injury, solidifying LPS’s suitability as an effective inducer for studying OS in pigs [6,10]. However, despite its established mechanistic role and utility in porcine models, limited studies have systematically evaluated the effect of LPS with different concentrations in pigs.

In reality, it is of great significance to investigate the effect of LPS in pigs, which provides not only more prevention strategies for OS but also a more representative animal model than rodents in human medical research due to the similarities between human beings and pigs, including anatomical size and structure, physiology, immunology and genomes [11,12]. In brief, OS is a critical factor that can significantly impact the health and productivity of pigs [13,14], making its study essential in the field of animal science and agriculture. Additionally, pigs serve as valuable models for human health research, as they share many physiological similarities with humans. Insights gained from studying OS in pigs can also contribute to a broader understanding of OS-related diseases in humans, offering potential benefits for both animal and human health [15,16]. Therefore, conducting research on oxidative stress in pigs is not only vital for the pig industry but also holds significant implications for comparative medicine and biomedical research.

This study aimed to characterize the dynamic progression of LPS-induced OS in WMP piglets and elucidate its temporal interplay with inflammatory responses, intestinal barrier integrity, metabolic profiles and gut microbiota composition. These findings will contribute to guarding against OS in young animals and offer the possibility of prevention strategies with microbiota. Meanwhile, this study will provide dosage reference for the OS model of pigs.

## 2. Materials and Methods

### 2.1. Materials

Lipopolysaccharide (LPS, CAS: 93572-42-0; purity > 97%) was purchased from Sigma-Aldrich (Shanghai) Trading Co., Ltd. (Shanghai, China); NaCl (CAS: 7647-14-5; purity ≥ 99%) was purchased from Xilong Scientific Co., Ltd. (Shantou, China). All solutions in this study were diluted and configured using deionized water, if not otherwise specified.

### 2.2. Animals and Experimental Design

Twenty-eight castrated male piglets of the Wuzhishan pig species (35 days old, 4.17 ± 0.46 kg) were obtained from Hainan Rural Animal Husbandry Ecological Agriculture Development Co., Ltd. (Qiongzhong, China) and were used as experimental animals. Throughout the study, the piglets were housed in pens at Nanhua Breeding Professional Cooperative (Ding’an, China) with 3–4 individuals per pen. Commercial creep feed was provided ad libitum in detachable feed troughs (Beijing Dabeinong Group Co., Ltd., Beijing, China), and water was freely accessible via nipple-type drinkers. The pens and residual feed were cleaned daily at 09:00.

After a 7-day acclimatization period, the piglets were randomly allocated to four treatments of seven replicates, each with one piglet per replicate. The treatments were as follows: (1) control group (control): intraperitoneal administration of sterile saline; (2) low-dose LPS group (L-LPS): intraperitoneal administration of LPS at 50 μg/kg body weight; (3) medium-dose LPS group (M-LPS): intraperitoneal administration of LPS at 100 μg/kg body weight; and (4) high-dose LPS group (H-LPS): intraperitoneal administration of LPS at 150 μg/kg body weight. The LPS dosing regimen was established through preliminary experiments, which determined both the minimum effective dose (50 μg/kg) required to elicit oxidative stress responses in piglets and the maximum tolerated dose (150 μg/kg). All piglets received equivalent volumes of LPS solution or sterile saline on day 0. On day 7, euthanasia was performed following the Guidelines for Experimental Animal Welfare [17]. Specifically, piglets were sequentially euthanized at 15 min intervals using electrical stunning followed by exsanguination and evisceration.

### 2.3. Sample Collection

Blood samples were collected from the precaval vein on day 0, day 1, day 3, day 5 and day 7. These samples were centrifuged at 3000 r/min for 10 min at 4 °C to obtain serum, and the serum was stored at −20 °C for further analysis.

Feces were collected daily after cleaning pens within 4 h, were snap-frozen in liquid nitrogen and stored at −80 °C for further analysis.

The duodenum, jejunum, ileum, colon, liver and spleen were treated with 4% paraformaldehyde, and the small intestines were treated with 2.5% glutaraldehyde in another sample. Briefly, the intestine was isolated from the abdominal cavity and distinguished. Then, all intestines had a complete segment (about 1 cm) cut out from their middle parts, respectively. These intestinal segments were stored in 4% paraformaldehyde solution at a volume 5–7 times that of the intestines. The liver and spleen samples were cut into 1 cm^3^ pieces and stored using the same method as the intestines. A 3 mm × 3 mm longitudinal section piece was excised from each segment intestine and stored in 2.5% glutaraldehyde solution at 4 °C, with the volume of the solution being 5–7 multiples of that of the intestine piece.

### 2.4. Serum Antioxidant Capacity, Inflammation-Related Cytokines and Intestinal Permeability Indexes

Serum levels of malondialdehyde (MDA, CAS: A003-1-2), total superoxide dismutase (T-SOD, CAS: A001-1-2), total antioxidant capacity (T-AOC, CAS: A015-1-2), catalase (CAT, CAS: A007-1-1), nitric oxide (NO, CAS: A012-1-2) and inducible nitric oxide synthase (iNOS, CAS: H372-1) were detected using colorimetry. The determination methods were conducted in strict line with the instructions of the Nanjing Jiancheng Biological kit (Nanjing, China). Inflammation-related cytokines, including interleukin-1β (IL-1β, CAS: ml025973), interleukin-6 (IL-6, CAS: ml025981) and tumor necrosis factor-α (TNF-α, CAS: ml002360) were measured according to the protocols of the Shanghai Enzyme-linked Biotechnology Co., Ltd. (Shanghai, China). Diamine oxidase (DAO, CAS: G0154W) and D-lactic acid (D-LA, CAS: G0827W) were detected using colorimetry according to the protocols of Suzhou Grace Biotechnology Co., Ltd. (Suzhou, China).

### 2.5. Histomorphology and Scanning Electron Microscopy

The intestines, livers and spleens were removed from the piglets and fixed with 4% paraformaldehyde. Meanwhile, the small intestines were fixed with 2.5% glutaraldehyde. All samples were sent to the Servicebio (Wuhan, China) for Hematoxylin-Eosin (H&E) staining or the production of scanning electron microscope (SEM) samples.

### 2.6. Serum Metabolomics Analysis

#### 2.6.1. Metabolites Extraction

A 50 μL aliquot of each sample was transferred to an Eppendorf tube. Subsequently, 200 μL of extraction solution was added (acetonitrile:methanol = 1:1, containing an isotopically labeled internal standard mixture). The samples were vortexed for 30 s, sonicated for 10 min in an ice-water bath and incubated for 1 h at −20 °C to precipitate proteins. Then the sample was centrifuged at 12,000 rpm (RCF = 13,800× *g*, R = 8.6 cm) for 15 min at 4 °C. The resulting supernatant was transferred to a fresh glass vial for analysis. The quality control (QC) sample was prepared by mixing an equal aliquot of the supernatants from all of the samples.

#### 2.6.2. LC-MS/MS Analysis

LC-MS/MS analyses were conducted using an ultra-high-performance liquid chromatography (UHPLC) system (Vanquish, Thermo Fisher Scientific, Waltham, MA, USA) equipped with a UPLC BEH Amide column (2.1 mm × 100 mm, 1.7 μm) and coupled with an Orbitrap Exploris 120 mass spectrometer (Orbitrap MS, Thermo). The mobile phase was composed of 25 mmol/L ammonium acetate and 25 mmol/L ammonia hydroxide in water (pH = 9.75) (A) and acetonitrile (B). The autosampler temperature was maintained at 4 °C, and the injection volume was set to 2 μL.

The Orbitrap Exploris 120 mass spectrometer was employed for its capacity to acquire MS/MS spectra in information-dependent acquisition (IDA) mode, controlled by the acquisition software (Xcalibur, v4.3, Thermo). In this mode, the acquisition software continuously evaluates the full scan MS spectrum. The electrospray ionization (ESI) source settings were as follows: sheath gas flow rate at 50 Arb, auxiliary gas flow rate at 15 Arb, capillary temperature at 320 °C, full MS resolution at 60,000, MS/MS resolution at 15,000, collision energy at 10/30/60 in NCE mode and spray voltage at 3.8 kV (positive) or −3.4 kV (negative).

#### 2.6.3. Data Preprocessing and Annotation

The raw data were converted to the mzXML format using ProteoWizard and processed with an in-house program, which was developed using R (v4.3.2) and which is based on XCMS, for peak detection, extraction, alignment and integration. Then an in-house MS2 database (BiotreeDB v3.0) was applied in metabolite annotation. The cutoff for annotation was set at 0.3.

### 2.7. Metagenomics

#### 2.7.1. DNA Extraction

DNA was extracted from various samples using the E.Z.N.A.^®^ Stool DNA Kit (CAS: D4015-02, Omega, Inc., Norcross, GA, USA) following the manufacturer’s protocol. This reagent, designed for extracting DNA from trace amounts of samples, has proven effective for preparing DNA from most bacterial species. Sample blanks, consisting of unused swabs processed through the DNA extraction procedure, were tested to ensure the absence of DNA amplicons. The total DNA was eluted in 50 µL of elution buffer, following a modified procedure based on the manufacturer’s instructions (QIAGEN), and was stored at −80 °C until PCR analysis.

#### 2.7.2. DNA Library Construction

The DNA library was constructed using the TruSeq Nano DNA LT Library Preparation Kit (CAS: FC-121-4001). DNA was fragmented using dsDNA Fragmentase (CAS: M0348S, NEB) by incubating at 37 °C for 30 min. The library construction process began with fragmented cDNA. Blunt-end DNA fragments were generated using a combination of fill-in reactions and exonuclease activity, followed by size selection with the provided sample purification beads. An A-base was added to the blunt ends of each strand to prepare them for ligation to the indexed adapters. Each adapter, containing a T-base overhang, was ligated to the A-tailed fragmented DNA. These adapters included the full complement of sequencing primer hybridization sites for single, paired-end and indexed reads. Single- or dual-index adapters were ligated to the fragments, and the ligated products were amplified via PCR under the following conditions: initial denaturation at 95 °C for 3 min; 8 cycles of denaturation at 98 °C for 15 s, annealing at 60 °C for 15 s, and extension at 72 °C for 30 s; followed by a final extension at 72 °C for 5 min.

#### 2.7.3. Data Analysis

Raw sequencing reads were processed to obtain valid reads for further analysis. Initially, sequencing adapters were removed from the reads using cutadapt (version 1.9). Subsequently, low-quality reads were trimmed using fqtrim (version 0.94) with a sliding-window algorithm. The reads were then aligned to the host genome using bowtie2 (version 2.2.0) to eliminate host contamination. Once quality-filtered reads were obtained, they were de novo assembled to construct the metagenome for each sample using IDBA-UD (version 1.1.1). All coding regions (CDS) of the metagenomic contigs were predicted using MetaGeneMark (version 3.26). CDS sequences from all samples were clustered using CD-HIT (version 4.6.1) to obtain unigenes. The abundance of unigenes for a specific sample was estimated by TPM, based on the number of aligned reads using bowtie2 (version 2.2.0). The lowest common ancestor taxonomy of unigenes was determined by aligning them against the NCBI NR database using DIAMOND (version 0.9.14). Similarly, functional annotations of unigenes were obtained. Differential analysis was conducted at each taxonomic, functional or gene-wise level using the Kruskal–Wallis test, based on the taxonomic and functional annotations of unigenes, along with their abundance profiles.

### 2.8. Statistical Analysis

All measurements were carried out with at least three replicates. Data were expressed as Mean with SDs. The Shapiro–Wilk test was used to assess the normality distribution. Statistical significance was determined using one-way ANOVA, followed by Duncan’s multiple comparisons in SPSS 26.0 (IBM, New York, NY, USA). Spearman correlation coefficients were calculated to quantify the degree of linear relationship among microbiota, metabolites and serum indexes. All results with a *p*-value < 0.05 were defined as significantly different.

## 3. Results

### 3.1. LPS Reduces Body Weight Gain and Elevates Oxidative Stress Parameters in Piglets

All of the piglets’ weights were recorded daily (as seen in Appendix A). The results showed that LPS resulted in the gain in weight to be low and even negative. However, there are no significant differences among the groups (*p* > 0.05). Table 1 shows serum oxidative stress parameters in piglets. The results suggest that the levels of T-AOC, T-SOD and CAT in the M-LPS and H-LPS groups were lower than those in the control group on day 1 (*p* < 0.05). Meanwhile, these parameters on day 1 were lower than these on day 0 and day 7 in the M-LPS and H-LPS groups (*p* < 0.05) and this similar alteration in the L-LPS group was only in CAT (*p* < 0.05). In addition, T-AOC and T-SOD levels of the H-LPS group were significantly decreased in comparison with the control group on day 3 and day 5 (*p* < 0.05) and T-AOC and CAT levels of the M-LPS group were also significantly decreased in contrast to the control group on day 3 (*p* < 0.05). MDA levels of the L-LPS, M-LPS groups and the H-LPS group were higher than that of the control group on day 1 (*p* < 0.05). In the L-LPS, M-LPS and H-LPS groups, compared to day 0, day 3, day 5 and day 7, higher MDA levels were observed on day 1 (*p* < 0.05).

### 3.2. LPS Increased Serum Inflammation-Related Mediators and Cytokines in Piglets

As shown in Table 2, the levels of iNOS and NO on day 1 showed differences between the control, M-LPS and H-LPS groups (*p* < 0.05) and the iNOS level on day 1 in M-LPS was higher than on day 0, day 3, day 5 and day 7 (*p* < 0.05). On day 1, IL-6 and IL-1β showed the highest levels in the M-LPS and L-LPS groups, when compared to other time points (*p* < 0.05). Additionally, the levels of TNF-α, IL-6 and IL-1β in the M-LPS and L-LPS groups were higher than those of the control group on day 1 (*p* < 0.05).

### 3.3. Effect of LPS on Intestinal Permeability, Morphology and Immune-Related Organs

The higher DAO and D-LA levels in the M-LPS and H-LPS groups than those in the control group were observed on day 1 (*p* < 0.05). In comparison with day 1, day 0, day 5 and day 7 showed the lower DAO and D-LA levels in the M-LPS and H-LPS groups (*p* < 0.05) (Table 3). Meanwhile, the morphological structure of the intestine was destroyed to varying degrees (Appendix A). In the M-LPS and H-LPS groups, lesions were also found in the livers and spleens, such as a larger hepatic sinus crevice, vacuolation of hepatocytes and inflammatory cell infiltration (Appendix A).

### 3.4. Analysis of Piglet Gut Microbiota

Based on the results of the above serum indicators, we selected the fecal or colon content samples of the control and M-LPS groups on day 1 and day 7 for microbiota analysis. On day 1, Appendix A showed that there was no difference in α-diversity between the control and M-LPS groups (*p* > 0.05). However, observed_species, Shannon, Simpson and Chao1 of the M-LPS group were higher than those of the control group on day 7 (*p* < 0.05) (Appendix A). From the β-diversity analysis, the principal component analysis (PCA) and principal coordinate analysis (PCoA) results showed specific differences in fecal or colonic microbiota on both day 1 and day 7 between the two groups (Figure 1).

We further performed analysis for microbial compositions, which were presented with explicit classification and a relative abundance greater than 0.5%. As seen in Appendix A, 7 phyla, 21 genera and 8 species were observed on day 1 and Spirochaetes, Tenericutes, *Bacteroides*, *Mycoplasma*, *Treponema* and *Oscillibacter_sp._CAG:241* showed significant differences between the two groups (*p* < 0.05). On day 7, 6 phyla, 13 genera and 11 species were found and Bacteroidetes, Firmicutes, Proteobacteria, Actinobacteria, Spirochaetes, Euryarchaeota, *Clostridium*, *Collinsella*, *Oscillibacter*, *Ruminococcus*, *Eubacterium*, *Blautia*, *Lachnoclostridium*, *Treponema*, *Bacteroides_thetaiotaomicron*, *Lactobacillus_amylovorus* and *Prevotella_sp._P5-92* showed significant differences between the two groups (*p* < 0.05) (Appendix A).

Figure 2A,B show the differential microbiota of the linear discriminant analysis (LDA) score ≥ 3 on day 1 and day 7 between the two groups. Both on day 1 and on day 7, more microbes increased in the M-LPS group than decreased ones. According to differential microbial unigenes (Figure 2C), we found that the unigenes were primarily enriched by microbial metabolism in diverse environments and biosynthesis of amino acids on day 1 (Figure 2D), but were primarily enriched by peptidoglycan biosynthesis, mismatch repair, galactose metabolism and DNA replication on day 7 (Figure 2E).

### 3.5. Analysis of Piglet Serum Metabolite

The serum samples of piglets of day 1 and day 7 in the control and M-LPS groups were detected by LC-MS/MS in the negative ion mode (ESI^−^) (Appendix A) and positive ion mode (ESI^+^) (Appendix A), respectively. To investigate the differential metabolites between the control and M-LPS groups, the orthogonal partial least squares-discriminant analysis (OPLS-DA) model was used to identify the serum metabolic profile. The OPLS-DA score plots indicated clear clustering trends of metabolites between the control and M-LPS groups in both the ESI^−^ mode and the ESI^+^ mode, suggesting significant changes in serum metabolism induced by LPS.

As seen in Appendix A, validation with 200 random permutation tests generated intercepts R^2^Y = 0.96, Q^2^ = −0.42 in the ESI^−^ mode, R^2^Y = 0.93 and Q^2^ = −0.51 in the ESI^+^ mode for the control versus M-LPS groups on day 1. Analogously, R^2^Y = 0.96, Q^2^ = −0.03 in the ESI^−^ mode, R^2^Y = 0.95 and Q^2^ = −0.03 in the ESI^+^ mode for the control versus M-LPS groups on day 7 were observed as shown in Appendix A. These results manifested that the OPLS-DA models had good robustness and there was no overfitting problem.

According to variable importance in the projection (VIP) > 1 and *p* < 0.05, we found a total of 72 significantly differential serum metabolites, 20 metabolites in the ESI^−^ mode and 52 metabolites in the ESI^+^ mode, respectively. Metabolites including hepoxilin B3, cohibin C, creatine and (9xi, 10xi, 12xi)-9,10-dihydroxy-12-octadecenoic acid were significantly increased in the M-LPS group on day 1, while gravacridonediolacetate, 2-oxoarginine, glyceric acid, guanidoacetic acid, isoleucyl-tryptophan, β-D-glucosamine, glycyl-valine, 5-aminoimidazole ribonucleotide, glycerophosphocholine, 4-chloro-3,5-dimethoxybenzyl alcohol, histamine, methylpyrazine, 3-methyl sulfolene, L-alanine, 1-butanethiol and β-D-3-[5-deoxy-5-(dimethylarsinyl)ribofuranosyloxy]-2-hydroxy-1-propanesulfonic acid were significantly decreased in the M-LPS group on day 1 (Figure 3A,B). On day 7, 45 significantly differential metabolites were found, with 16 metabolites in the ESI^−^ mode and 29 metabolites in the ESI^+^ mode, respectively. Differential metabolites including urocanic acid, isonicotinic acid, cohibin C, gingerglycolipid B and dimethylethanolamine were significantly increased in the M-LPS group, while thymine, thymine, fexofenadine, phenobarbital, brassicanal A, serylphenylalanine, lepidine F, histamine and glycyl-valine were significantly decreased in the M-LPS group (Figure 3C,D).

The complex metabolic reactions and their regulation in organisms are not performed alone, but complex different genes and proteins form complex pathways and networks, and mutual influence and regulation eventually lead to systematic changes of the metabolome. In the present study, most of the differential metabolites were enriched by glycine, serine and threonine metabolism in the ESI^−^ and ESI^+^ modes. Additionally, a wealth of differential metabolites were related to amino-acid metabolism, including phenylalanine metabolism, histidine metabolism, alanine, aspartate and glutamate metabolism, arginine and proline metabolism and d-glutamine and d-glutamate metabolism (Figure 4A,B). Meanwhile, most of differential metabolites between the control and M-LPS groups on day 7 were also enriched in histidine metabolism, alanine, aspartate and glutamate metabolism and cysteine and methionine metabolism (Figure 4C,D).

### 3.6. Spearman Correlation Analysis

Further, Spearman correlation analysis was performed on all samples from the control and M-LPS groups to explore relationships among the differential microbiota, differential metabolites and serum indicators. The differential microorganisms were not considered for correlation analysis when the fold change varied between 0.5 and 2 though differences were significant. Likewise, the differential metabolites were performed for correlation analysis with the fold change > 4 or <0.25, VIP > 1 and *p* < 0.05.

As shown in Figure 5A, on day 1, *Oscillibacter_sp._CAG:241* had significant positive correlations with lepidine F, T-AOC, isoleucyl-tryptophan, T-SOD and glycine and significant negative correlations with MDA, IL-6, IL-1β, TNF-α and iNOS (*p* < 0.05). Lepidine F had significant positive correlations with T-AOC, T-SOD and *Mycoplasma* and significant negative correlations with MDA, IL-6, IL-1β, TNF-α, iNOS and DAO (*p* < 0.05). D-LA had significant negative correlations with T-AOC, isoleucyl-tryptophan, T-SOD, CAT, Tenericutes and glycine and significant positive correlations with MDA, IL-6, creatine, IL-1β, hepoxilin B3 and *Treponema* (*p* < 0.05). DAO had significant negative correlations with isoleucyl-tryptophan, T-SOD, CAT, Tenericutes and *Mycoplasma* and significant positive correlations with IL-1β, hepoxilin B3, TNF-α, iNOS and NO (*p* < 0.05).

Figure 5B shows the correlation analysis between differential metabolites and microbiota. *Lactobacillus_amylovorus* was significantly negatively correlated with *Prevotella_sp._P5-92*, *Clostridium*, *Blautia* and Euryarchaeota and significantly positively correlated with Bacteroidetes and *Collinsella* (*p* < 0.05). Lepidine F had significant positive correlations with Actinobacteria and *Collinsella* and significant negative correlations with Euryarchaeota, *Blautia*, *Firmicutes_bacterium_CAG:110*, *Ruminococcus*, Spirochaetes, *Lachnoclostridium*, *Eubacterium* and *Prevotella_sp._P5-92* (*p* < 0.05).

## 4. Discussion

LPS, a component of the outer wall of Gram-negative bacterial cell wall and as a common endotoxin, can stimulate activation immune responses in multiple organs and tissues. In bovine mammary epithelial cells, LPS promoted inflammatory genes and proteins by inciting NF-κB and MAPK, but the Nrf2 pathway was inhibited due to the crosstalk between the NF-κB and Nrf2 pathways [8,18]. Meanwhile, it was found that LPS increased NOX2/ROS/NF-κB pathway expression and the level of inflammatory cytokines, including IL-1β, IL-6 and TNF-α in lung epithelial A549 cells [9]. Not surprisingly, the Nrf2 signaling pathway was inhibited by LPS in lung epithelial A549 cells [10]. In addition, LPS resulted in varying degrees of injury and inflammation in macrophages, osteoblast, Caco-2 cells, liver and spleen [19,20,21,22,23]. In the present study, LPS treatments increased the cytokines’ secretion (IL-6, IL-1β and TNF-α) with the downstream of the NF-κB signaling pathway. These cytokines presented the highest level on day 1 and dropped in the following days until they became close to the control group suggesting the occurrence of systemic inflammation in the body on day 1. More importantly, IL-1β and TNF-α, two endogenous pyrogens, provoke fever by directly participating in the synthesis and release of certain central mediators of fever such as prostaglandin E2 or indirectly stimulating mediators in cortical cells of the brain [24,25]. Additionally, a previous study indicated that the increase of NO and iNOS play critical roles in endotoxin-induced fever as well [26]. Thus, the elevated temperature and the disruption of appetite signaling receptors by LPS may be responsible for the decrease of body weight in LPS treatments [27,28].

The crosstalk between NF-κB and Nrf2 pathways is pivotal in linking inflammation to OS. LPS-induced activation of NF-κB not only promotes the secretion of pro-inflammatory cytokines (IL-6, IL-1β, TNF-α) but also inhibits Nrf2-mediated transcription of antioxidant genes [10,15], directly contributing to the reduced activity of T-SOD and CAT (downstream targets of Nrf2) in the M-LPS and H-LPS groups on day 1. This impairment in antioxidant defenses, paired with the concurrent accumulation of MDA (a lipid peroxidation end product) and NO (a pro-oxidative factor), forms a “pro-inflammatory-pro-oxidative” positive feedback loop—where oxidative damage products further amplify NF-κB activation [18]. This loop explains the synchronous peak of OS markers on day 1: T-AOC, T-SOD and CAT were significantly reduced, while MDA, NO and iNOS were markedly elevated, coinciding with the highest levels of pro-inflammatory cytokines. The inverse correlation between antioxidant enzyme activity and MDA levels further validates oxidative damage secondary to this dysregulated pathway crosstalk [29]. Notably, this OS response was transient but intense: by day 7, most OS markers (T-AOC, T-SOD, CAT, MDA, NO, iNOS) largely recovered to control levels, reflecting restored systemic redox homeostasis. However, histopathological damage in the intestine, liver and spleen persisted, indicating residual oxidative injury despite the resolution of acute OS markers.

The systemic inflammation and OS developed a challenge for gut health with the increase in intestine permeability and the decrease in the tight junction protein. DAO, an intracellular enzyme existing in mammalian intestinal mucosa or ciliated epithelial cells, catalyzes the oxidation of diamine possessing a protective effect on intestinal mucosa. However, DAO could enter the circulatory system in the case of impaired intestine barrier function; it, therefore, acted as a biomarker of gut permeability [30]. Though D-LA serves as a biomarker of gut permeability as well, it mainly comes from various bacteria in the gut that break down sugars. In physiological conditions, high concentrations of D-LA do not occur in the mammalian circulatory system due to the lack of enzymes to break it down, but D-LA can enter the circulation when the intestinal barrier function is injured [31]. Here, we observed that the levels of DAO and D-LA in the serum of piglets were significantly elevated within one day after LPS treatment. The possible reason is that the reduced activity of T-SOD and CAT on day 1 impaired the clearance of reactive oxygen species (ROS) in intestinal epithelial cells, leading to lipid peroxidation (elevated MDA) and subsequent damage to tight junction proteins (e.g., occludin, claudin) [30]. These findings suggest that LPS treatment induces intestinal damage, which was further corroborated by H&E staining and scanning electron microscopy of the intestinal tissue. Moreover, on the whole, the piglets were treated with the higher concentration LPS, which had the longer recovery time.

To comprehensively assess the effect of acute oxidative stress and inflammatory response of LPS-challenge on piglets, we performed metagenomics analysis to identify the microbial composition between the control and M-LPS groups on day 1 and day 7. A strain, *Oscillibacter_sp._CAG:241*, with a clearly defined classification was found after conditional screening. The relative abundance of *Oscillibacter_sp._CAG:241* was significantly lower in the M-LPS group compared to the controls on day 1. Notably, Spearman correlation analysis revealed that *Oscillibacter_sp._CAG:241* abundance was positively associated with serum antioxidant markers and negatively correlated with pro-inflammatory cytokines (IL-6, IL-1β, TNF-α) and oxidative damage markers (MDA). While direct evidence of its antioxidant function is limited, these associations suggest that *Oscillibacter_sp._CAG:241* may buffer OS by enhancing antioxidant capacity or suppressing pro-oxidative factors, though its mechanistic role in redox homeostasis remains unexplored. In the previous report, it was believed that the association between *Oscillibacter_sp._CAG:241* and insulin sensitivity was statistically significant due to correlations manifesting significant correlation with glucose metabolism [32]. Interestingly, researchers found that the relative abundance of *Oscillibacter_sp._CAG:241* was significantly positively correlated with the plasma level of LPS [33]. The diversity of bacterial metabolic enzyme functions in various micro-environment to underpin the collective metabolism of microbiomes may be responsible for these differences [34]. Furthermore, little is known for more functions of *Oscillibacter_sp._CAG:241*; it is of great necessity to investigate the role of this strain in OS.

Four strains, *Bacteroides_thetaiotaomicron*, *Lactobacillus_amylovorus*, *Firmicutes_bacterium_CAG:110* and *Prevotella_sp._P5-92* demonstrated significant differences between the control and M-LPS groups on day 7. It was reported that nonalcoholic fatty liver disease (NAFLD) with liver inflammation and injury could be ameliorated due to *Bacteroides_thetaiotaomicron* administration and regulating gut microbial composition, enhancing gut-liver folate and unsaturated fatty acid metabolism, which were responsible for the alleviation of metabolic syndrome [35]. In addition, *Bacteroides_thetaiotaomicron*, as a gut commensal bacterium, is capable of generating bacterial extracellular vesicles that elicited NF-κB activation in the presence of TLR4 and Toll-interleukin-1 receptor domain-containing adaptor protein. Notably, both cell-type and health status influence the communication between the vesicles and host [36] and lipids of the vesicles’ decreased *Shigella flexneri* virulence [37]. However, *Bacteroides_thetaiotaomicron* could have significant pro-inflammatory effects in the presence of *Bilophila wadsworthia* [38]. The latter was not observed in our study.

It was deemed that *Lactobacillus* boasts antioxidative potential in ameliorating D-galactose-induced aging [39] and *Lactobacillus_amylovorus* was widely reported in improving intestinal function through producing enzymes, regulating microflora and upregulating tight junction protein [40,41]. Of note, *Lactobacillus_amylovorus* grew preferentially with peptide-rich rather than amino acid-rich substrates [42], which may be answerable for enriched amino acid metabolism of differential metabolites in serum on day 7. Although the significant alterations of serum indicators were presented on day 1, the microbial differences were more pronounced on day 7. To our best knowledge, the microbial response to OS lags behind the body’s response, which might be verified by the previous report with host damage before “opportunistic fungus” altered it [43]. The different sampling positions, where feces were obtained on day 1 and colon content were sampled on day 7, could also be a cause. *Oscillibacter_sp._CAG:241*, *Bacteroides_thetaiotaomicron* and *Lactobacillus_amylovorus* were decreased in the M-LPS group suggesting a potential for oxidative injury, and their metabolites may play a critical role.

Here, we performed non-targeted metabolomics to identify key differential metabolites. A signature metabolite, lepidine F, was found both on day 1 and day 7. Lepidine F belongs to the class of organic compounds known as biphenyls and derivatives according to the description from the Human Metabolome Datebase. Unfortunately, it was rarely reported in previous studies. However, it is still an undeniable fact that this metabolite was closely related to OS. Lepidine F is a consistently decreased metabolite in the M-LPS group, and it showed strong positive correlations with T-AOC and T-SOD, alongside negative correlations with MDA, pro-inflammatory cytokines (IL-6, IL-1β, TNF-α), iNOS, and DAO. This indicates lepidine F may act as a potential mediator of redox balance, linking microbial activity to OS alleviation and gut-barrier protection. Meanwhile, it had a significant positive correlation with potential probiotics such as *Oscillibacter_sp._CAG:241*, and a significant negative correlation with potentially harmful bacteria, including *Firmicutes_bacterium_CAG:110* and *Prevotella_sp._P5-92*. On the contrary, hepoxilin B3 negatively related to lepidine F might contribute to lung inflammations [44]. It seems unsurprising that the expression of this metabolite was increased by systemic inflammation LPS-induced. Moreover, the results showed a significant positive correlation between hepoxilin B3 and pro-inflammatory cytokines. Hepoxilin B3 has also been identified among the downregulated metabolites following the amelioration of rheumatoid arthritis, which may further elucidate its pro-inflammatory potential [45]. In diabetes, steroid hormone metabolism could be influenced by increased hepoxilin B3 levels [46]. Thus, the close relationship between steroid and lipids metabolism may be responsible for enrichment in glycerolipid and glycerophospholipid metabolism.

The antioxidant properties of proteins and peptides mainly depend on the types and amount of amino acids (AAs). Herein, many AAs were decreased in the M-LPS group, which not only blocked biosynthesis of antioxidant enzymes but was answerable for the enriched AA metabolism of differential metabolites. A growing number of studies have revealed that the microbiota AA metabolism influences the host AA content in the circulatory system, suggesting that more AAs were utilized by microbes in OS [47], which was verified by microbial unigenes’ enrichment in microbial metabolism in diverse environments and biosynthesis of AAs. More importantly, AAs were harnessed by immune cells to achieve rapid multiplication and serve as energy source precursors in the TCA cycle in response to inflammation [48], which resulted in the decrease of AAs in circulation. These results suggested an inflammatory response following the LPS injection.

Creatine is an organic compound that occurs naturally in human muscle tissue synthesized primarily in the liver, kidneys and pancreas. In the current study, we found that creatine was increased in the M-LPS group. In earlier studies, it was believed that creatine acts as an important role in cellular metabolism, particularly during metabolically stressed states, and limitations in the availability for transport and/or storing creatine can weaken metabolism [49]. The increase of creatine in the tissue could alleviate the severity of injury. Therefore, increasing the creatine level may be a way for piglets to self-regulate in response to inflammation and OS. Additionally, a fever response caused by endogenous pyrogens (TNF-α, IL-1β) is essentially an increased energy consumption result [25], and creatine can be converted to phosphocreatine to provide energy. On day 7, although creatine was also detected, no significant difference was found between the control and M-LPS groups. This alteration was similar to the level of pro-inflammatory cytokines and the activity of antioxidant enzymes, which indicated the most significant shift on day 1 and then gradually returned to the physiological level. However, it is of great importance to realize that microbial response to an adverse environment may lag behind the host’s own response in a short time. Intestinal inflammation is a major problem in pups. After the extremely high concentration of LPS (10 mg/kg body weight) treatment for 12 h, LPS-induced injury of the piglet colon was observed [50], but it is difficult to give rise to significant differences in gut microbiota with such a short period. In the current study, though we screened a bacterial strain, *Oscillibacter_sp._CAG:241*, with anti-inflammation and anti-oxidant potential on day 1, it is of great necessity to further investigate its role in alleviating piglet gut inflammation. Collectively, the results in our study showed that the severity of inflammation and OS tended to be self-regulated within 7 days over time, but it is noted that microbial alterations might be inconsistent with this trend.

## 5. Conclusions

To conclude, this study establishes a LPS-induced OS model in WMP piglets, with 100 μg/kg body weight identified as the optimal dose. Key OS markers—suppressed antioxidant capacity (T-AOC, T-SOD, CAT) and elevated oxidative damage (MDA, NO, iNOS)—peaked at day 1, recovering systemically by day 7. Critically, persistent histopathological damage in the intestine, liver and spleen confirmed sustained OS-induced tissue injury, validating the model’s physiological relevance. Associated inflammatory responses, amino acid metabolism alterations and microbiota perturbations (e.g., depleted *Oscillibacter_sp.CAG:241*) were temporally linked to OS progression. Observed inflammatory responses, amino acid metabolism alterations, and microbiota changes were temporally linked to OS progression. This standardized model provides a robust tool for OS research using 100 μg/kg LPS in porcine studies.

## Figures and Tables

**Figure 1 vetsci-12-00694-f001:**
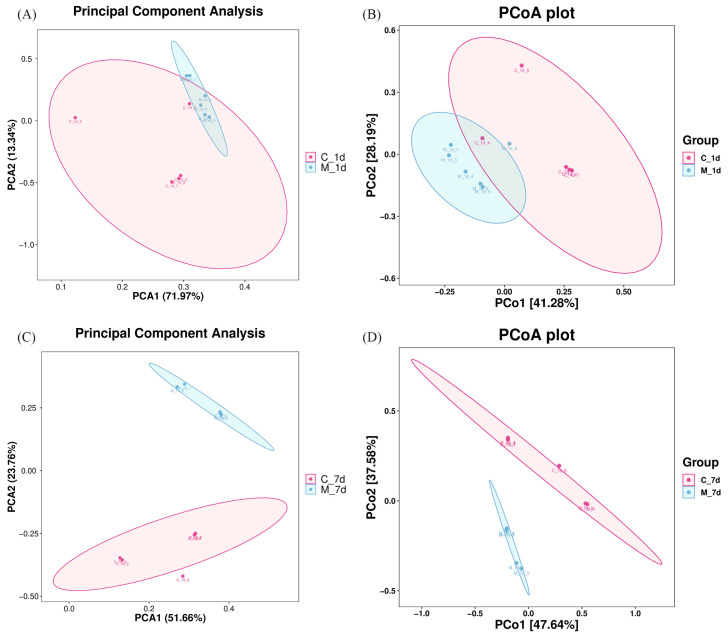
Effects of LPS on β-diversity of piglet gut microbiota. (**A**,**B**) PCA and PCoA of fecal microbial communities in the control and M-LPS groups on 1 d; (**C**,**D**) PCA and PCoA of colonic microbial communities in the control and M-LPS groups on 7 d.

**Figure 2 vetsci-12-00694-f002:**
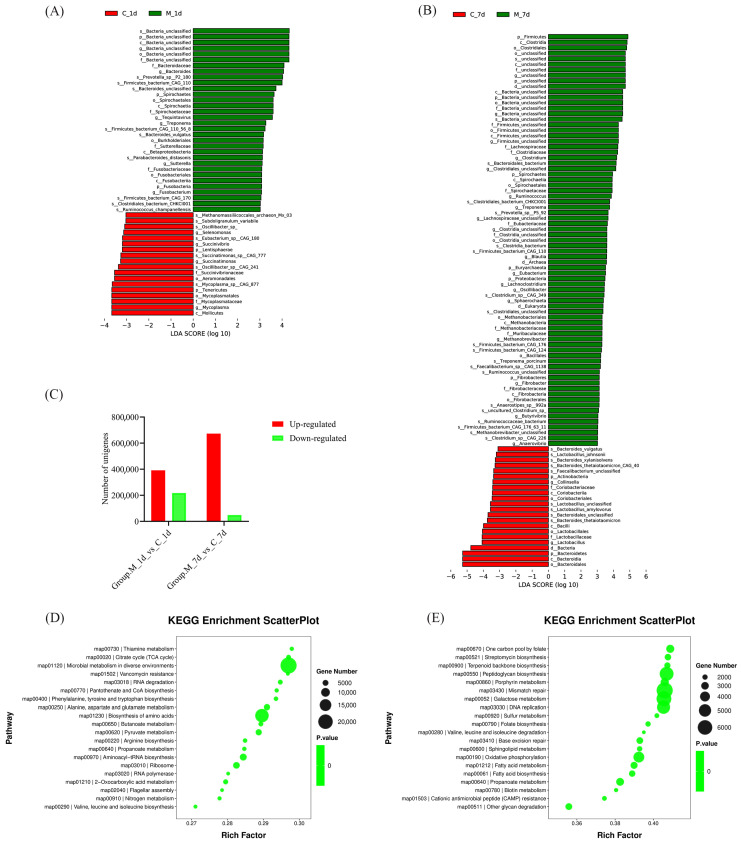
Effects of LPS on the microbial composition and function of piglet gut microbiota. (**A**) LDA effect size (LEfSe) of key microbial taxa that contribute to structural differences in fecal microbiota on 1 d and (**B**) colonic microbiota on 7 d between the control and M-LPS groups. Only taxa meeting an LDA threshold ≥ 3 were shown; (**C**) differential microbial unigenes of fecal and colon content between the control and M-LPS groups on 1 d and 7 d; (**D**) KEGG enrichment scatter plot of differential fecal microbial unigenes on 1 d and (**E**) colonic microbial unigenes on 7 d.

**Figure 3 vetsci-12-00694-f003:**
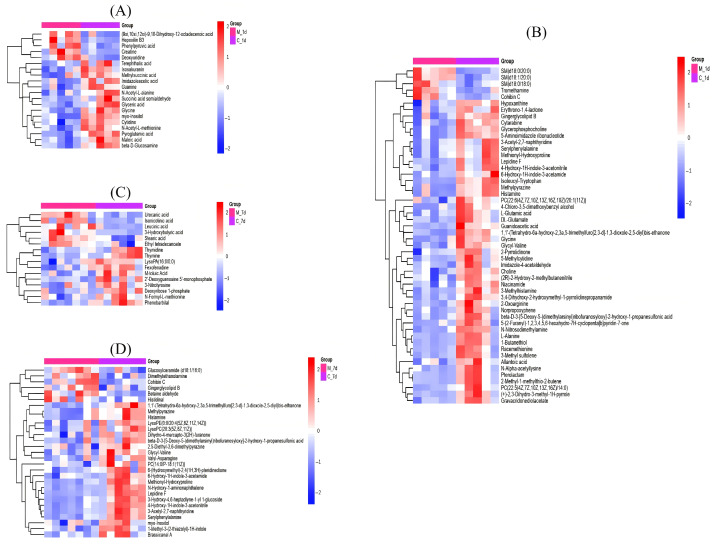
Heatmap of differentially enriched serum metabolites between the control and M-LPS groups. (**A**) In ESI^−^ mode (1 d); (**B**) in ESI^+^ mode (1 d); (**C**) in ESI^−^ mode (7 d); (**D**) in ESI^+^ mode (7 d).

**Figure 4 vetsci-12-00694-f004:**
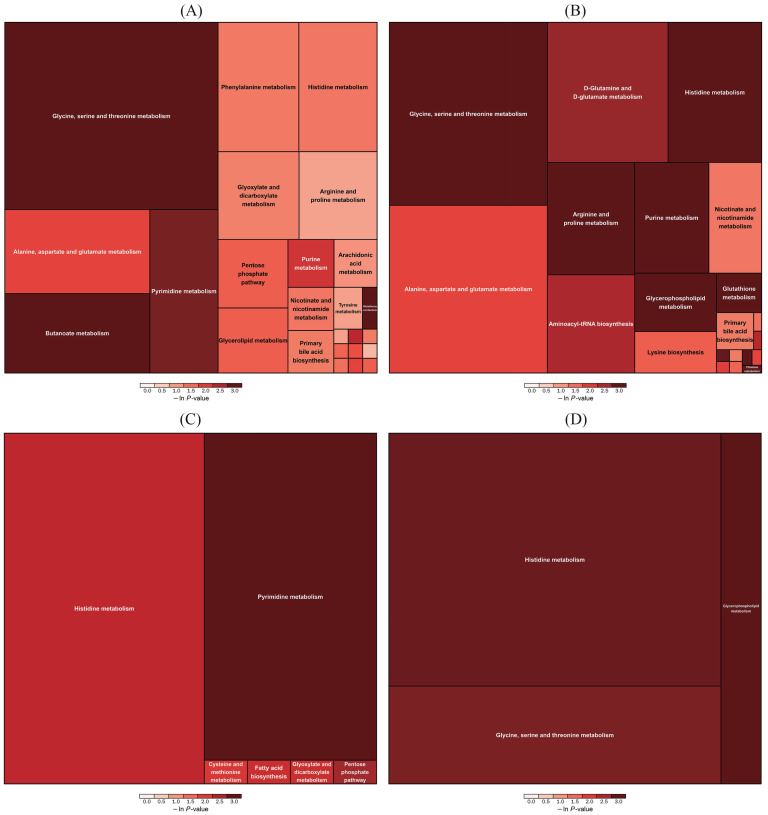
KEGG pathways analysis of differential serum metabolites between the control and M-LPS groups. (**A**) In ESI^−^ mode (1 d); (**B**) in ESI^+^ mode (1 d); (**C**) in ESI^−^ mode (7 d); (**D**) in ESI^+^ mode (7 d).

**Figure 5 vetsci-12-00694-f005:**
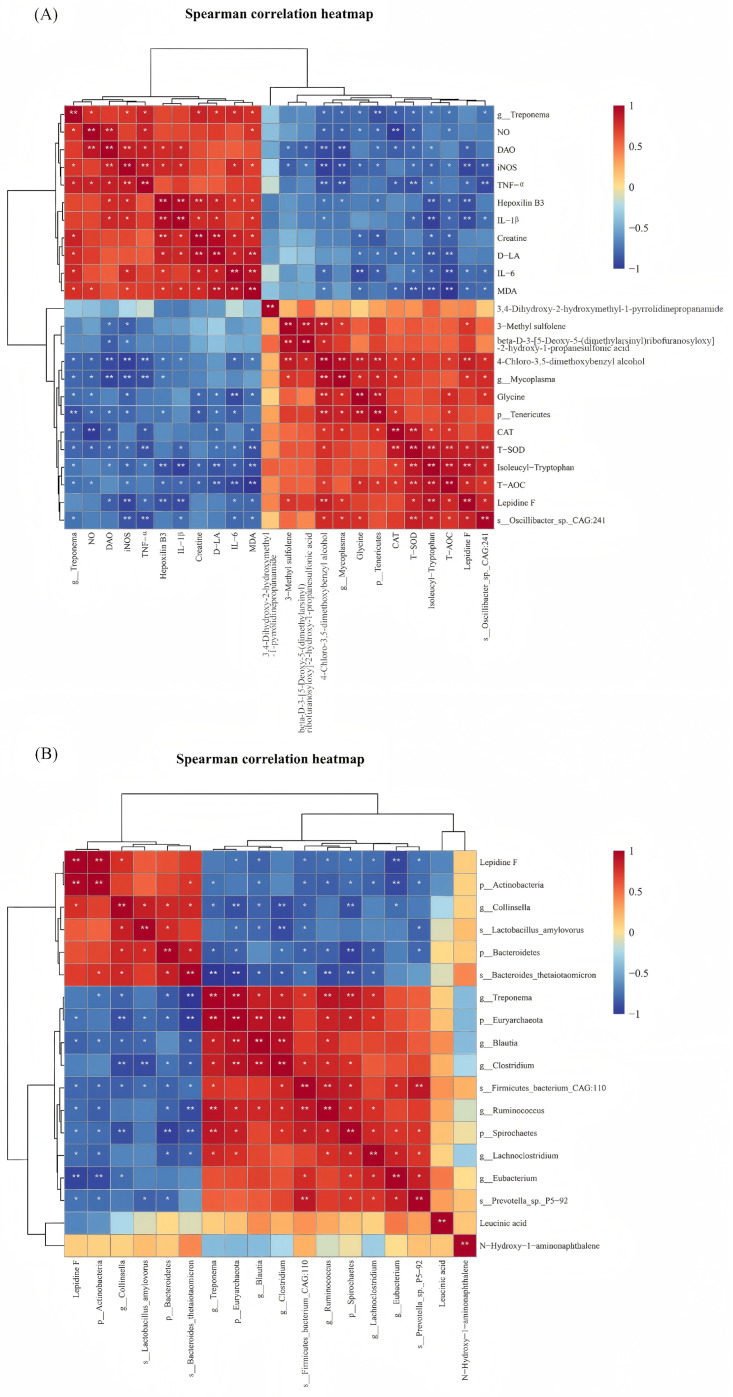
Spearman correlation heatmap. (**A**) Spearman correlation analysis among the differential microbiota, differential metabolites and serum indicators; (**B**) Spearman correlation analysis between differential metabolites and microbiota. * *p* < 0.05, ** *p* < 0.01.

**Table 1 vetsci-12-00694-t001:** Effects of LPS on serum oxidative stress parameters in piglets.

Items		Groups			
		Control	L-LPS	M-LPS	H-LPS
T-AOC (U/mol)	Day 0	3.05 ± 0.60	3.08 ± 1.36	3.00 ± 1.25 ^A^	3.30 ± 1.50 ^A^
	Day 1	3.13 ± 0.70 ^a^	2.40 ± 1.22 ^b^	1.29 ± 0.40 ^Cc^	0.89 ± 0.37 ^Cc^
	Day 3	2.66 ± 0.67 ^a^	2.30 ± 0.77 ^a^	1.53 ± 0.60 ^Cb^	1.69 ± 0.24 ^BCb^
	Day 5	2.77 ± 0.88 ^a^	2.22 ± 0.75 ^ab^	1.92 ± 0.60 ^BCb^	1.97 ± 0.99 ^BCb^
	Day 7	2.67 ± 0.61	2.34 ± 0.63	2.40 ± 0.88 ^AB^	2.07 ± 1.58 ^B^
T-SOD (U/mL)	Day 0	107.20 ± 7.50	105.10 ± 16.83	103.34 ± 7.59 ^A^	103.88 ± 26.58 ^A^
	Day 1	106.60 ± 12.60 ^a^	98.50 ± 24.95 ^ab^	90.44 ± 8.33 ^Bb^	88.02 ± 10.15 ^Bb^
	Day 3	109.89 ± 12.69 ^a^	107.74 ± 11.41 ^a^	102.47 ± 10.37 ^Aab^	96.34 ± 15.36 ^ABb^
	Day 5	109.25 ± 3.22 ^a^	109.51 ± 5.23 ^ab^	101.84 ± 13.50 ^Ab^	104.94 ± 17.65 ^Ab^
	Day 7	115.51 ± 19.98	107.77 ± 13.92	105.01 ± 9.26 ^A^	109.33 ± 20.09 ^A^
CAT (U/mol)	Day 0	23.59 ± 1.35	23.21 ± 2.25 ^A^	22.89 ± 1.40 ^A^	22.99 ± 1.70 ^A^
	Day 1	22.00 ± 5.32 ^a^	16.46 ± 6.68 ^Bab^	14.62 ± 8.91 ^Bb^	11.73 ± 7.28 ^Cb^
	Day 3	23.78 ± 1.67 ^a^	21.56 ± 3.63 ^Ab^	20.68 ± 3.29 ^Ab^	19.37 ± 2.95 ^Bb^
	Day 5	23.28 ± 2.27	22.22 ± 3.08 ^A^	22.34 ± 1.91 ^A^	22.22 ± 2.04 ^AB^
	Day 7	23.15 ± 1.18	22.27 ± 2.98 ^A^	22.44 ± 1.18 ^A^	22.50 ± 1.37 ^A^
MDA (mmol/mL)	Day 0	7.63 ± 0.42 ^B^	7.53 ± 0.27 ^B^	7.44 ± 0.51 ^B^	7.52 ± 0.71 ^B^
	Day 1	7.98 ± 0.30 ^Ab^	8.52 ± 0.28 ^Aa^	8.64 ± 1.01 ^Aa^	8.67 ± 0.70 ^Aa^
	Day 3	7.80 ± 0.39 ^AB^	7.78 ± 0.41 ^B^	7.75 ± 0.40 ^B^	7.70 ± 0.40 ^B^
	Day 5	7.71 ± 0.48 ^AB^	7.55 ± 0.56 ^B^	7.61 ± 0.89 ^B^	7.95 ± 0.89 ^B^
	Day 7	7.71 ± 0.35 ^AB^	7.54 ± 0.32 ^B^	7.46 ± 0.46 ^B^	7.65 ± 0.60 ^B^

Note: Different lowercase letters in the same row indicate significant differences; different uppercase letters in the same column indicate significant differences (n = 3–7).

**Table 2 vetsci-12-00694-t002:** Effects of LPS on serum inflammation-related mediators and cytokines in piglets.

Items		Groups			
		Control	L-LPS	M-LPS	H-LPS
NO (μmol/L)	Day 0	16.24 ± 5.05	17.97 ± 9.10	16.32 ± 11.46	17.39 ± 9.91 ^C^
	Day 1	16.35 ± 11.99 ^c^	25.64 ± 13.68 ^bc^	28.85 ± 8.73 ^b^	39.86 ± 11.14 ^Aa^
	Day 3	22.30 ± 8.52 ^b^	21.40 ± 9.83 ^b^	21.21 ± 9.61 ^b^	32.72 ± 9.63 ^ABa^
	Day 5	24.37 ± 16.68	25.30 ± 15.96	25.91 ± 17.98	23.55 ± 14.8 ^BC^
	Day 7	18.05 ± 15.56	23.62 ± 16.68	20.30 ± 12.98	22.11 ± 12.17 ^BC^
iNOS (μmol/mL)	Day 0	10.77 ± 1.75	10.99 ± 1.81	11.04 ± 1.08 ^B^	12.44 ± 2.02
	Day 1	9.70 ± 1.76 ^b^	10.93 ± 2.60 ^ab^	12.95 ± 3.45 ^Aa^	12.96 ± 3.44 ^a^
	Day 3	10.08 ± 2.74	11.95 ± 4.75	11.72 ± 2.85 ^AB^	11.38 ± 4.62
	Day 5	9.79 ± 2.63	10.57 ± 4.39	11.78 ± 2.87 ^AB^	11.76 ± 3.19
	Day 7	9.65 ± 3.79	9.57 ± 5.16	11.28 ± 2.72 ^AB^	10.17 ± 4.47
IL-6 (pg/mol)	Day 0	21.42 ± 9.41	18.52 ± 10.91 ^B^	21.79 ± 12.86 ^B^	24.98 ± 12.51 ^C^
	Day 1	22.10 ± 8.89 ^b^	40.52 ± 7.09 ^Aa^	49.88 ± 14.75 ^Aa^	52.65 ± 5.69 ^Aa^
	Day 3	20.93 ± 5.51 ^b^	30.29 ± 14.65 ^ABab^	32.87 ± 12.44 ^Bab^	36.10 ± 4.97 ^Ba^
	Day 5	22.53 ± 4.40	24.40 ± 8.75 ^B^	25.27 ± 9.48 ^B^	27.93 ± 6.39 ^BC^
	Day 7	15.69 ± 7.19	15.64 ± 12.92 ^B^	25.45 ± 13.25 ^B^	27.03 ± 5.23 ^BC^
IL-1β (pg/mol)	Day 0	8.84 ± 6.02	9.56 ± 8.73 ^B^	10.04 ± 5.22 ^B^	12.73 ± 5.92 ^C^
	Day 1	11.26 ± 5.40 ^c^	23.45 ± 5.76 ^Ab^	31.20 ± 9.20 ^Aab^	37.93 ± 10.61 ^Aa^
	Day 3	12.52 ± 8.24 ^b^	11.91 ± 2.84 ^Bb^	16.87 ± 7.72 ^Bab^	24.28 ± 10.87 ^Ba^
	Day 5	12.14 ± 8.48	8.65 ± 6.84 ^B^	15.85 ± 5.37 ^B^	10.32 ± 7.14 ^C^
	Day 7	14.49 ± 12.83	16.82 ± 14.1 ^AB^	17.97 ± 7.85 ^B^	11.28 ± 8.40 ^C^
TNF-α (pg/mol)	Day 0	40.12 ± 6.78	38.42 ± 11.09	43.91 ± 11.79 ^B^	42.89 ± 6.53 ^B^
	Day 1	38.08 ± 12.21 ^b^	43.43 ± 12.13 ^b^	67.70 ± 12.41 ^Aa^	73.43 ± 14.38 ^Aa^
	Day 3	33.53 ± 6.08 ^b^	36.68 ± 10.09 ^ab^	41.81 ± 13.67 ^Bab^	47.59 ± 7.80 ^Ba^
	Day 5	35.38 ± 11.44	31.88 ± 6.96	34.21 ± 14.35 ^B^	46.01 ± 13.19 ^B^
	Day 7	38.74 ± 8.54	36.12 ± 13.35	38.07 ± 12.35 ^B^	39.88 ± 14.79 ^B^

Notes: Different lowercase letters in the same row indicate significant differences; different uppercase letters in the same column indicate significant differences (n = 3–7).

**Table 3 vetsci-12-00694-t003:** Effects of LPS on intestinal permeability in piglets.

Items		Groups			
		Control	L-LPS	M-LPS	H-LPS
DAO (U/mL)	Day 0	34.22 ± 1.52	34.77 ± 1.94 ^AB^	32.59 ± 1.72 ^B^	32.93 ± 3.23 ^E^
	Day 1	35.77 ± 6.74 ^b^	40.93 ± 4.05 ^Ab^	58.04 ± 10.36 ^Aa^	58.88 ± 3.57 ^Aa^
	Day 3	32.56 ± 4.04 ^c^	34.26 ± 1.04 ^Bc^	40.07 ± 3.89 ^Bb^	54.78 ± 2.35 ^Ba^
	Day 5	30.69 ± 3.76 ^b^	33.35 ± 7.65 ^ABb^	35.16 ± 3.02 ^Bb^	36.75 ± 1.51 ^Ca^
	Day 7	31.99 ± 1.01	34.82 ± 6.47 ^AB^	36.50 ± 4.77 ^B^	37.89 ± 1.77 ^D^
D-LA (μmol/L)	Day 0	235.72 ± 14.19	244.05 ± 18.03 ^AB^	244.87 ± 26.51 ^B^	242.06 ± 24.34 ^B^
	Day 1	222.06 ± 19.16 ^c^	274.04 ± 17.67 ^Ab^	297.91 ± 27.62 ^Ab^	344.90 ± 21.41 ^Aa^
	Day 3	246.60 ± 13.16 ^b^	243.18 ± 8.24 ^ABb^	262.95 ± 21.01 ^Bb^	333.78 ± 23.18 ^Aa^
	Day 5	244.24 ± 31.45 ^ab^	238.69 ± 18.57 ^Bb^	241.77 ± 11.53 ^Bab^	250.70 ± 11.06 ^Ba^
	Day 7	237.60 ± 29.30	235.55 ± 24.90 ^B^	236.04 ± 27.88 ^B^	245.12 ± 29.05 ^B^

Note: Different lowercase letters in the same row indicate significant differences; different uppercase letters in the same column indicate significant differences (n = 3–7).

## Data Availability

The original contributions presented in the study are included in the article; further inquiries can be directed to the corresponding author.

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
