# Peer review of "Oxidative Stress Model of Lipopolysaccharide-Challenge in Piglets of Wuzhishan Miniature Pig"

_vetsci, 2025, doi:10.3390/vetsci12080694_

Round 1
Reviewer 1 Report
Comments and Suggestions for Authors
Through the complexity of the analyses and the analytical interpretation, compared to a considerable number of bibliographic sources, your work constitutes an important source of information in the field.
As you shown, the Lipopolysaccharide (LPS) found in the outer membrane of Gram-negative bacteria, possesses immune-activating properties, being used to establish inflammation models due to its potent proinflammatory effects. Your study demonstrates that the lipopolysaccharide determines general inflammatory response, metabolism affecting and microbiota perturbation, the inflammatory response was the most severe on day 1 and gradually recovered until close to the level of the control group. In high levels, LPS treatment induces intestinal damage, increasing of intestine permeability and the decrease of tight junction protein, and the changing of microbial composition, the bacterial metabolic enzymes determining lesions in guts, spleen and liver disease, such as larger hepatic sinus crevice, vacuolation of hepatocytes and inflammatory cell infiltration similar to a nonalcoholic fatty disease.
Author Response
We sincerely appreciate your meticulous review and valuable comments on our manuscript. Your approval of the work has greatly encouraged us, and we deeply recognize that this is an important affirmation of our research efforts.
Reviewer 2 Report
Comments and Suggestions for Authors
This study aimed to evaluate the effects of different concentrations of lipopolysaccharide on serum parameters, intestinal morphology, and inflammation in Wuzhishan Miniature Pig piglets, and further explored critical bacteria and metabolites related to anti-oxidative stress. The paper is well written, the results are exposed, and the discussion supports the results. A few suggestions are reported below:
Introduction: Please extend the information regarding the correlation between lipopolysaccharide and oxidative stress.
M&M: lines 98-104. Discuss of different groups treated with different treatments.
Results: Table 2. No treatments, but groups.
All figures are not readable. Modify them.
Reviewer 3 Report
Comments and Suggestions for Authors
The article is interesting and supplies a detailed response regarding the Wuzhishan pig's interaction with LPS. In this study, the authors aim to systematically investigate the dynamic effects and temporal trends of OS induced with LPS over time.
The response of pigs to LPS is well established, particularly in terms of inflammatory and oxidative stress markers, as well as gut microbiota analysis. However, the rationale for using LPS primarily as a model for oxidative stress is not entirely clear, given that oxidative stress typically arises as a consequence of inflammation. In the Discussion section, greater emphasis should be placed on oxidative stress markers and their relationship with other measured parameters. Additionally, although correlations were performed with OSI markers, these were not actually commented. Overall, the tone of the conclusion appears somewhat misaligned with the stated objectives of the study (see comment below).
To strengthen the study’s coherence, the authors should either refine the aim to better reflect the broader scope of their investigation OR place greater emphasis on the oxidative stress aspects—both in results and interpretation.
Introduction
The introduction is consistent with the title and objectives but it does not maintain coherence with the later discussion and conclusions drawn.
Conclusion
While the study sets out to characterize an LPS-induced oxidative stress model in Wuzhishan piglets, the conclusions focus more heavily on inflammation, metabolic changes, and microbiota alterations than on oxidative stress itself. As a result, the conclusions seem somewhat misaligned with the primary aim.
Minor issues
Results
Line 237: Table 1. In the table you also have a marker of oxidative stress - MDA, so please change the table title.
Line 345: Whether a Spearman Correlation Analysis was performed on the entire sample. Add that part
Discussion
423-424 “The relative abundance of Oscillibacter_sp._CAG:241 in the control group was higher than that in the M-LPS group suggesting its antioxidant or function probiotics potential”
Expand the section related to the potential antioxidant effect and support with a reference. I think that there are no studies confirming that Oscillibacter sp. has a clearly defined antioxidant function – it is a hypothesis based on correlations, not proven mechanisms.
Round 2
Reviewer 2 Report
Comments and Suggestions for Authors
Thank you for the revisions required.
Reviewer 3 Report
Comments and Suggestions for Authors
The authors satisfactorily addressed all my concerns in the revised version of the manuscript. Therefore, I have no further comments.